# Individual and Combined Effects of Paternal Deprivation and Developmental Exposure to Firemaster 550 on Socio-Emotional Behavior in Prairie Voles

**DOI:** 10.3390/toxics10050268

**Published:** 2022-05-22

**Authors:** Sagi Enicole A. Gillera, William P. Marinello, Mason A. Nelson, Brian M. Horman, Heather B. Patisaul

**Affiliations:** 1Department of Biological Sciences, NC State University, Raleigh, NC 27695, USA; sagiller@ncsu.edu (S.E.A.G.); wpmarine@ncsu.edu (W.P.M.); mnelson7@ncsu.edu (M.A.N.); bmhorman@ncsu.edu (B.M.H.); 2Center for Human Health and the Environment, NC State University, Raleigh, NC 27695, USA

**Keywords:** endocrine disruptors, flame retardants, sexual differentiation, pair bond

## Abstract

The prevalence of neurodevelopmental disorders (NDDs) is rapidly rising, suggesting a confluence of environmental factors that are likely contributing, including developmental exposure to environmental contaminants. Unfortunately, chemical exposures and social stressors frequently occur simultaneously in many communities, yet very few studies have sought to establish the combined effects on neurodevelopment or behavior. Social deficits are common to many NDDs, and we and others have shown that exposure to the chemical flame retardant mixture, Firemaster 550 (FM 550), or paternal deprivation impairs social behavior and neural function. Here, we used a spontaneously prosocial animal model, the prairie vole (Microtus ochrogaster), to explore the effects of perinatal chemical (FM 550) exposure alone or in combination with an early life stressor (paternal absence) on prosocial behavior. Dams were exposed to vehicle (sesame oil) or 1000 µg FM 550 orally via food treats from conception through weaning and the paternal absence groups were generated by removing the sires the day after birth. Adult offspring of both sexes were then subjected to open-field, sociability, and a partner preference test. Paternal deprivation (PD)-related effects included increased anxiety, decreased sociability, and impaired pair-bonding in both sexes. FM 550 effects include heightened anxiety and partner preference in females but reduced partner preference in males. The combination of FM 550 exposure and PD did not exacerbate any behaviors in either sex except for distance traveled by females in the partner preference test and, to a lesser extent, time spent with, and the number of visits to the non-social stimulus by males in the sociability test. FM 550 ameliorated the impacts of parental deprivation on partner preference behaviors in both sexes. This study is significant because it provides evidence that chemical and social stressors can have unique behavioral effects that differ by sex but may not produce worse outcomes in combination.

## 1. Introduction

Extensive epidemiological and animal-based work has demonstrated that exposure to early life stressors (ELS) during critical windows of development spanning pregnancy to late adolescence can exacerbate susceptibility to adult disease [1,2], including a higher risk of socio-emotional disorders. Globally, the prevalence of neurodevelopmental disorders (NDDs), many with a strong sex bias, is inexplicably rising. The causal factors are most certainly multi-faceted, but the rapidity of the increase implicates environmental factors as primary drivers. Chemical exposures are widely thought to be major contributors to NDD risk. Ample epidemiological data have linked developmental exposure to air pollution, pesticides, lead, flame retardants (FRs), and other pollutants with a higher risk of autism spectrum disorders (ASD), attention deficit hyperactivity disorder (ADHD), cognitive deficits, and other impairments [3,4,5,6]. Direct experimental evidence is lacking linking specific chemicals to adverse effects on socio-emotional behaviors common to many NDDs. Moreover, nearly nothing is known about how other environmental insults might exacerbate the damage carried out in the developing brain due to chemical exposures, but the available evidence suggests it is likely to be underestimated [7,8,9,10,11]. Using a uniquely suitable animal model, the prairie vole (*Microtus ochrogaster*), here we explored how socio-emotional behaviors are impacted by early life chemical exposure alone (FRs), and in combination with ELS (paternal deprivation (PD)).

That complex environmental exposures from various sources can affect a person’s health is now well recognized. The totality of exposures a person experiences from conception to death is referred to as the “exposome”, a concept that has become increasingly important for understanding the environmental causes of disease [12]. The exposome includes ELS known to impair pre- or post-natal human brain development. These include social stressors such as poor parental care, bullying, social isolation, and sexual assault [13,14,15]. In addition, children raised in poverty (>20% of US children and dramatically higher elsewhere) are at the greatest risk of combined chemical exposure and ELS, and also have higher rates of cognitive and behavioral disorders [16,17,18]. Thus, the present studies address the pressing need to understand how chemical exposures might adversely impact the brain and behavior in a more human-relevant context.

The precipitous increase in NDDs has prompted some to label it a silent pandemic [7,19,20]. Child mental health problems are strongly associated with persistent social function problems, lower academic success, higher risk of suicide, and adult mental health problems [21]. Anxiety disorders are by the far the most common psychiatric disorder in US children, affecting nearly 33% between 13 and 18 with a median age of onset of age 11. Girls have higher anxiety and depression rates than boys and, in both sexes, the age of onset is getting progressively younger. By contrast, social disorders, such as ASD, are roughly four times more prevalent in boys, and girls present with different sets of symptoms and deficits [22]. Thus, mental health disorders manifest in each sex differently and can persist for a lifetime. Because they are so common to NDDs and have a strong sex bias, here we focused on anxiety- and affiliation-related behaviors.

FRs were selected as the chemical class of interest because human exposure is extensive, they can bioaccumulate, and were associated with adverse cognitive and behavioral outcomes [23,24,25], including a heightened risk of ASD [6,7,26,27,28,29,30,31]. There are multiple FR classes, with the brominated FRs (BFRs) being the most intensively studied. The use of one subgroup, the polybrominated diphenyl ethers (PBDEs), has largely been phased out after being linked to thyroid disruption and cognitive impairment [23,31]. Thus alternatives are being phased in, including newer brominated forms and organophosphate ester FRs (OPFRs) [32]. Compelling work by the National Toxicology Program [33,34], along with experimental and epidemiological data [35,36,37,38,39,40], suggest they too have the potential to be developmentally neurotoxic, prompting some to label them “regrettable substitutes” [41,42]. Because organophosphate esters used as FRs were used in other applications dating back decades, human exposure also dates back decades and has been increasing over time [43].

Here, we used the commercial mixture Firemaster 550 (FM 550), which is often applied to foam-based products including furniture, strollers, and car seats. FM 550 is composed of two brominated compounds [2-ethylhexyl-2,3,4,-5-tetrabromobenzoate (EH-TBB) and bis(2-ethylhexyl) 2,3,4,5-tetrabromophthalate (BEH-TEBP)] the organophosphate triphenyl phosphate (TPP) and several isopropylated triarylphosphate isomers (ITPs) [44,45]. FM 550 components or metabolites are routinely detected in human tissues including breast milk, blood, hair, fingernails and urine [46,47,48] with children having higher levels than adults [49,50,51,52]. Compounding evidence from us and others shows that FM 550 can adversely impact socio-emotional behavior and neurodevelopment [33,35,53,54,55,56,57,58,59], and epidemiological studies have linked maternal exposure to FM 550 or its components with impaired cognitive and behavioral performance in children [60,61]. Finally, our lab has also repeatedly shown in rodents that developmental exposure to FM 550 or its components alters, in a sexually dimorphic manner, anxiety-like [55,56,59] and socio-emotional behaviors, including pair-bonding in the socially monogamous species, the prairie vole [57,58].

The ELS used for the present studies was PD because poor paternal care or absence increases the risk of numerous negative outcomes associated with social and emotional health [62], including increased risk of psychopathology [63], substance abuse, delinquency, and susceptibility to other diseases [64,65]. It was robustly demonstrated that perinatal social stress, broadly, can adversely impact offspring’s behavioral development and conduct [66,67,68,69]. Children raised in stressful environments characterized by conflict, separation, and low attachment are more predisposed to behavioral problems [62]. Previous studies in multiple bi-parental species also report a variety of negative outcomes associated with a lack of paternal care [70]. Bi-parental care (BPC) is rare in mammals and only seen in about 3–5% of species. Laboratory strains of mice and rats do not naturally display BPC, hence we used prairie voles, which are a uniquely valuable animal model for studying BPC, affiliation, and pair-bonding [71,72,73].

Prairie vole dams and sires have similar parental responsibilities. Father behavior includes retrieving and grooming the pups, carrying food to older offspring, providing pups with warmth (huddling), and defending the nest, all of which can occur in the mother’s absence [72,74,75]. Prairie vole partners coordinate their parenting behaviors and if the father is removed, the dams do not display parental compensation such as increased huddling, nursing, or foraging [73]. Thus, offspring of single mothers have less total parental contact time, which is considered an ELS. Paternal absence was shown to alter offspring’s socio-emotional behavior, including parental care [76,77]. For example, female offspring display impaired pair-bonding and both sexes participate in less licking and grooming of their own pups.

Critically, we already showed that this species is vulnerable to chemical exposures, including FM 550 [54,57,78], demonstrating their unique utility for investigating the effects of combined environmental insults on offspring socio-emotional behavior [70,77]. We first demonstrated that developmental exposure to FM 550 heightened generalized anxiety in females resulting in aversion to novelty [57]. In a brief 10-min partner preference test (PPT), exposed females displayed heightened increased partner preference while exposed males did not display one at all [57]. We subsequently replicated these findings in a different cohort of prairie voles using a 3-hr PPT. Exposed females displayed a stronger partner preference across the entirety of the test than same-sex controls, while exposed males failed to display a partner preference as quickly or consistently as same-sex controls [58]. These studies provide consistent evidence of disrupted sociality following developmental FM 550 exposure and the utility of the prairie vole model for probing the impact of environmental stressors on prosocial behaviors.

To explore the effects of perinatal chemicals (FM 550 exposure) and ESL (paternal absence) on prosocial behavior, we exposed prairie vole dams to 1000 µg FM 550 orally via food treats beginning on the day of conception and ending at pup weaning. Thus, pup exposure was gestational and lactational via the dam. Dams were not dosed by individual weight, but rather by the average colony weight of 50 g during pregnancy, producing exposures of approximately 20 mg/kg BW. Dose selection was based on our prior behavioral studies in rats and voles [56,57], and because it is well below the presumed NOAEL of 50 mg/kg/bw per day for the BFR combination in FM 550 [79]. To generate the PD groups, sires were removed the day after birth. Once the offspring were adults, we assessed anxiety levels, exploratory behavior, and the ability to form a pair bond. We hypothesized that FM 550 and PD would have a compounding negative effect on both sexes, resulting in increased social deficits and decreased exploratory and affiliative behavior.

## 2. Materials and Methods

The ARRIVE (Animal Research: Reporting of In Vivo Experiments) Guidelines “Essential 10” Checklist for Reporting Animal Research was used in the construction of this manuscript with all elements met [80]. The ARRIVE guidelines were developed in consultation with the scientific community as part of an NC3Rs (National Centre for the Replacement Refinement and Reduction of Animals in Research) initiative to improve the standard of reporting of research using animals.

### 2.1. Animals

Animal care, maintenance and experimental protocols met the standards of the Animal Welfare Act and the U.S. Department of Health and Human Services ‘Guide for the Care and use of Laboratory Animals’ and were approved by the North Carolina State University (NCSU) Institutional Animal Care and Use Committee (IACUC). Prairie voles (*Microtus ochrogaster*) were obtained from founders generously gifted in 2017 by Bruce S. Cushing at the University of Texas-El Paso and bred in house as indicated in humidity- and temperature-controlled rooms at 22 °C and 30% average humidity, each with 12 h:12 h light:dark cycles (lights on at 6AM EST) in the Assessment and Accreditation of Laboratory Animal Care (AAALAC) approved Biological Resource Facility at NCSU. Food (Lab Diet 5326 high fiber rabbit diet, St. Louis, MO, USA) and water were provided ad libitum. As in our prior studies and in accordance with recommended practices for endocrine-disrupting chemical (EDC) research, all animals were housed in conditions specifically designed to minimize unintended EDC exposure including the use of glass water bottles with metal sippers, woodchip bedding, and thoroughly washed polysulfone caging. The diet is not a low phytoestrogen diet (content varies a lot) because high fiber and at least some phytoestrogen content are required to maximize health and fertility of this herbivorous species [81].

### 2.2. Dose Preparation

As we carried out previously [55,57,59], sesame oil-based dosing solutions were prepared and coded by the laboratory of Dr. Heather Stapleton at Duke University and transferred to the Patisaul lab at NCSU where dosing and subsequent testing were performed. Briefly, a commercial mixture of FM 550 from Great Lakes Chemical (West Lafayette, IN, USA) [82] was used to prepare the dosing solution (1000 µg/20 µL) by weighing the appropriate amount of FM 550 and diluting it in HPLC-grade sesame oil (Sigma, Burlington, MA, USA) with stirring for 6 h, and then stored in amber bottles at 4 °C until use. FM 550 doses were selected based on our prior work in voles in which FM 550 exposure altered anxiety-like and social behavior [57] and well below the purported NOAEL of 50 mg/kg/day for the BFR components (there is no published NOAEL for the OPFR components) [59]. A small aliquot of the mixture was analyzed by gas chromatography–mass spectrometry in the Stapleton lab to confirm doses were accurate prior to exposure.

### 2.3. Exposure

Voles mate for life and breed continuously. Accordingly, dams from the breeding colony were randomly assigned to a dose group and exposure occurred from the day after parturition of the previous litter, designated as gestational day (GD) 0, through weaning (PND 21). All pairs had reared prior litters and were experienced parents. Dosing occurred daily between 10:00–11:00 a.m. Dams were given 1/4 of a soy-free, highly palatable food treat pellet (chocolate-flavored AIN-76A Rodent Diet Test Tabs, Test Diet, Richmond, IN, USA) with 20 µL of the vehicle (sesame oil) or 20 µL containing 1000 µg FM 550 in solution as is routinely performed in our lab. Dams were monitored to ensure the entire treat was consumed. To generate the PD groups, sires were removed the day after birth. This route of exposure was selected to minimize handling stress [59,63] and because oral dosing is the primary human exposure route. The dam is the preferred statistical unit for toxicological studies to avoid litter effects, but the individual pups were used as the statistical unit for logistical reasons (this species produces small litters and must remain pair bonded, which limits the capacity to generate a large number of individual litters). Because the colony is wild-derived and thus considerably more genetically diverse than the typical mouse and rat strains, that potential confound was handled statistically by testing for litter effects using an ANCOVA with litter as a covariant to which none were identified. Because voles are continuous breeders, some pairs produced multiple experimental litters (either sequential biparental litters or a biparental then a PD litter) to generate a sufficient number of experimental offspring. Once a sire was removed to generate the PD litter, the sire was not returned to the home cage (they were either used for other experiments or euthanized) and the pair did not generate another experimental litter. To limit potential confounding effects of litter or lineage, pups were never obtained from more than two litters from the same pair, and a maximum of 5 total pups per sex total were selected per mated pair (in most cases the number was less). Breeding occurred continuously over a 6-month period with each litter randomly assigned to each experimental group. The number of dams represented by offspring sex in each group is: BPC control (8 F, 9 M), BPC FM 550 (7 F, 9 M), PD control (6 F, 7 M), 2000 μg FM 550 (7 F, 11 M).

### 2.4. Behavioral Testing

As in our prior prairie vole experiments [57,58,83], all behavioral testing was conducted in a room at the NCSU Biological Resources Facility specifically dedicated and equipped for this purpose. Lab-housed prairie voles are diurnal, thus, all testing was conducted under white light between 10:00 am and 3:30 pm and video recorded by a camera suspended overhead for later analysis using TopScan (Clever Sys Inc., Reston, VA, USA) software with no people or other distractions in the room. Behavioral testing occurred in either a high walled (43 cm) blue, open-field (OF) area (58 cm × 58 cm) or a 3-chambered arena made of plexiglass with a total length of 198.12 cm, 30.48 cm deep and 30.48 cm wide and divided into three chambers roughly equal in size with small compartments (17.78 × 30.48 × 30.48 cm) on either side. Wired cups were used to restrain the animals (Stoelting, Wood Dale, IL, USA, item number 60451). The room contained multiple arenas, but males and females were never tested in the room simultaneously. Males were tested first each day, and all equipment was thoroughly cleaned with ethanol and a peroxide-based cleaner between testings.

Behavioral studies began on PND 60, which is considered young adulthood and after sexual maturity, and conducted in a sequence considered least to most intensive for the animals: OF, sociability, then partner preference. Animals were tested in only one behavioral test per day and the full battery was completed no later than 4 months of age. For experiments using “strangers,” these animals were approximately the same size or smaller, and sexually naive. While used more than once, “strangers” were used for no more than 3 h per day and were monitored for signs of stress or distress. The overall study design is shown in Figure 1.

### 2.5. Open-Field Test

The open-field test (OF) tracks investigation of a novel environment and was used to examine anxiety and exploration in a variety of rodents [84], including prairie voles [85,86,87]. Adult prairie voles were subjected to a standard 30min OF test as described previously [55]. Briefly, the test animal was gently placed in the center of an empty open arena and allowed to explore for 30 min freely. The center was defined digitally by dividing the task floor into a 3 × 3 square grid of equal size using the TopScan software. Endpoints were distance traveled in the entire arena and latency, number of entries, and time in the center. All videos were individually reviewed by an observer blind to exposure to exclude any trials in which an error or some other erroneous factor occurred. Twenty-one test animals (11 females and 10 males) were excluded from OF analysis due to technical or other issues. Final animal numbers were as follows: BPC control (19 F, 25 M); BPC FM 550 (18 F, 17 M); PD control (15 F, 14 M); PD + FM 550 (18 F, 18 M).

### 2.6. Sociability Test

For the sociability test (ST), using a 3-chambered arena, the subject animal is given the opportunity to explore a novel “stranger” same-sex animal or an empty wired cup to access social and exploratory motivation, with prairie voles typically being prosocial, particularly males [88]. Two wired cups were positioned at opposite ends of the 3-chambered arena; one empty and one holding a same-sex stranger. The wired cup allows the animals to interact but not harm each other. All strangers were given time to acclimate to the restrainer cup prior to testing. The test animal was gently placed in the center of the middle chamber and given 10 min to explore. Time spent in contact and entries with the stranger cup or empty cup chamber, and total distance traveled were analyzed. A sociability index was calculated by subtracting time spent with the inanimate object from time spent with the stranger animal, divided by the total time spent with both. A sociability index of 1.0 indicates 100% preference for the stranger and −1.0 indicates 100% preference for the empty cup. Two test animals, both females, were excluded from ST analysis due to technical or other issues. Final animal numbers were as follows: BPC control (23 F, 28 M); BPC FM 550 (22 F, 20 M); PD control (15 F, 16 M); PD + FM 550 (19 F, 20 M).

### 2.7. Partner Preference Test

Prairie voles spontaneously display social monogamy and a partner preference test (PPT) assesses the strength of the bond [89,90]. Test animals were cohabitated for 24 h with an opposite sex, unrelated, sexually naïve “partner” of reproductive age. This length of time is sufficient to induce a pair bond even if mating does not occur [90]. As in the ST, two cups were positioned on opposite ends of the 3-chambered arena. One cup held the partner of the test animal and the other held a stranger animal. The stranger was a novel, opposite sex, unrelated, sexually naïve animal. The animals in the cups were given time to acclimate. Time spent in proximity to the stranger cup or partner cup was recorded. The test animal was gently placed in the middle chamber and given 2 hrs to explore because, in our prior, relevant study, we found a sufficient length for a partner preference to emerge [58]. To gain more resolution, PP data were also binned into 30-min intervals as performed previously [58]. A partner preference index was calculated by subtracting the time spent with the stranger animal from the time spent with the partner animal, divided by the total time spent with both. One female climbed on top of one cup and was therefore excluded. Final animal numbers were as follows: BPC control (18 F, 23 M), BPC FM 550 (19 F, 16 M), PD control (11 F, 15 M), PD + FM 550 (15 F, 18 M).

### 2.8. Statistical Analysis

Statistical analyses were performed using Graph Prism, version 9.3.1 (La Jolla, CA, USA). A ROUT outliers test (Q = 1%) was used to identify and remove statistical outliers. For all analyses, statistical significance was defined as α ≤ 0.05. A 3-way ANOVA was used to assess if there was a main effect of sex, exposure and paternal care or any significant interactions. For behavioral endpoints, unexposed males and females were compared using a Student’s one-tailed *t*-test to check for known and hypothesized sex differences, as we carried out previously for similar studies [55,91], with the identification of known sex differences interpreted as confirmation that the studies were sufficiently powered to detect biologically meaningful effect sizes. Within sex, a 2-way ANOVA was then used to determine if there was a main effect between sex and paternal care followed up with a Fisher’s LSD post-hoc test for multiple comparisons. For all tasks where an investigation index was calculated, animals identified as statistical outliers, or that had total investigation times less than 2% of the total testing time (and thus considered non-participatory) were excluded. The behavioral indices were analyzed by a one-sample Wilcoxon test to determine if they were significantly different from chance (index of 0.0). PP data were binned into 30-min intervals and analyzed by a two-way repeated-measures ANOVA and Fisher’s protected Least Significant Difference (LSD) test within sex. For the PPT, PPI was analyzed using a one-sample Wilcoxon test at each 30 min time interval to determine how preference changed over the course of the test. For each outcome, effect size was calculated as recommended by multiple behavioral neuroscience groups, including The American Psychological Association [92,93]. ANOVA effect size was determined by calculating Eta squared (η^2^) and partial Eta squared (η_p_^2^). Effects are defined as small at 0.01, medium at 0.06, and large at 0.14. *T*-test effect size was calculated by Cohen’s d which is defined as small at 0.2, medium at 0.5, and large at 0.81.

## 3. Results

### 3.1. Open-Field

OF data were analyzed using a three-way (Figure 2A) and two-way ANOVA within a sex (Figure 2B). No interaction between exposure and PD was found for OF behavior by three-way ANOVA with the main effects of paternal care, sex, and exposure (Figure 2A). Although there was no main effect of sex, expected sex differences in the unexposed controls reared by both parents were observed for the center duration (Figure 2C, t_42_ = 1.669, *p ≤* 0.05, d = 0.51), with females spending more time in the center than males. A main effect of paternal care was found for center entries (Figure 2B, F_1, 133_ = 6.230, *p* ≤ 0.01) and latency to enter the center (F_1, 123_ = 7.517, *p* ≤ 0.007), as well as a main effect of exposure on distance traveled (F_1, 123_ = 3.987, *p* ≤ 0.05).

Since OF behavior within control animals was sexually dimorphic, OF data were next analyzed within sex using a two-way ANOVA, and the significant effects were primarily in the males. In females, there were no significant main effects, although there were suggestive effects of an interaction between exposure and PD on time spent in the center (Figure 2B, F_1, 65_ = 3.077, *p* ≤ 0.08, η_p_^2^ = 0.05) and bouts of center entries (F_1, 66_ = 3.370, *p* ≤ 0.07, η_p_^2^ = 0.05). Follow-up pairwise comparisons using a Fisher’s LSD post-hoc affirmed that all three manipulated groups spent less time in the center (Figure 2C) and made fewer center entries (Figure 2D) than the BPC controls.

Within males, no interaction between exposure and PD was found (Figure 2B). There was a main effect of FM 550 on time spent in the center (Figure 2B, F_1, 68_ = 1.522, *p* ≤ 0.01, η_p_^2^ = 0.09) and distance traveled (Figure 2B, F_1, 67_ = 6.064, *p* ≤ 0.02, η_p_^2^ = 0.08) with both increasing with exposure. Additionally, a main effect of paternal care was observed on latency (Figure 2B, F_1, 66_ = 13.23, *p* ≤ 0.0005, η_p_^2^ = 0.17) and bouts (Figure 2B F_1, 67_ = 7.518, *p* ≤ 0.0008, η_p_^2^ = 0.10) with PD males taking longer to enter the center and making fewer entries than BPC males.

### 3.2. Sociability Test

The ST data set was first analyzed with a three-way ANOVA (Figure 3B), where the main effects of paternal care were found for distance traveled (Figure 3B, F_1, 149_ = 3.748, *p* ≤ 0.05) and bouts visiting the empty cup (F_1, 150_ = 2.724, *p* ≤ 0.10), in addition to numerous significant and potential interactions. There were no significant interactions between exposure and PD, but interactions between PD and sex were found for distance traveled (Figure 3B, F_1, 149_ = 0.2334, *p* ≤ 0.02) and bouts visiting the stranger (F_1, 150_ = 4.590, *p* ≤ 0.03). Thus, the three-way ANOVA revealed that the exploratory behaviors in this task (distance traveled and bouts in each chamber) were more impacted than the social ones.

In the two-way ANOVA (Figure 3C), PD was the only main effect for any outcome, but the significantly affected endpoints differed by sex. A main effect of PD was found in males for time spent with the empty cup (Figure 3C, F_1, 79_ = 5.940, *p* ≤ 0.02, η_p_^2^ = 0.07), with PD males spending a greater time (Figure 3F). Despite not reaching significance, there was a perceivable effect of paternal care on the sociability index (Figure 3D, F_1, 79_ = 3.668, *p* ≤ 0.06, η_p_^2^ = 0.04) and stranger duration (Figure 3E, F_1, 79_ = 3.424, *p* ≤ 0.07, η_p_^2^ = 0.04) in males, with PD reducing both. In females, there was a significant effect of PD on distance traveled (Figure 3G, F_1, 70_ = 7.344, *p* ≤ 0.009, η_p_^2^ = 0.09) and bouts visiting both the stranger (Figure 3H, F_1,71_ = 4.986, *p* ≤ 0.03, η_p_^2^ = 0.07) and the empty cup (Figure 3I, F_1, 70_ = 4.087, *p* ≤ 0.05, η_p_^2^ = 0.06), with all three elevated compared to BPC females. When comparing unexposed BPC controls only to test for baseline sex differences, a sex difference was only found for distance traveled, with males traveling more than females (Figure 3G, t_46_ = 1.980, *p* ≤ 0.05, d = 0.57).

In males, pairwise comparisons revealed that the combination of FM 550 exposure and PD resulted in increased bouts with (Figure 3F, *p* ≤ 0.04, d = 0.59) and time spent with the empty cup (Figure 3F, *p* ≤ 0.02, d = 0.64), and a trending lower sociability index (Figure 3D, *p* ≤ 0.07, d = 0.50) with a medium effect size compared to BPC control males. In females, pairwise comparisons revealed that only the PD group without exposure to FM 550 was statistically significant for the three exploratory outcomes compared to BPC controls (Figure 3G–I). The most compelling example was bouts visiting a stranger (Figure 3H, *p* ≤ 0.06, d = 0.63). The PD + FM 550 females were not different from the BPC females, which suggests that co-exposure to PD and FM 550 ameliorated the main PD effect to some degree. None of the female groups displayed a preference for either the stranger or the empty cup, although all tended toward the stranger (Figure 3D).

### 3.3. Partner Preference Test

Social (Figure 4) and exploratory (Figure 5) activity in the PPT was analyzed using a three-way ANOVA and were both found to be strongly sexually dimorphic. Sex was identified as a significant main effect in the three-way ANOVA for the partner preference index (Figure 4A, F_1, 126_ = 23.49, *p* ≤ 0.0001), partner duration (Figure 4A, F_1, 126_ = 21.04, *p* ≤ 0.0001), stranger duration (Figure 4A, F_1, 125_ = 24.35, *p* ≤ 0.0001), and visits to the stranger chamber (Figure 5A, F_1, 124_ = 4.559, *p* ≤ 0.03). Main effects of paternal care were found for all social endpoints (Figure 4B, PPI: F_1, 126_ = 7.488, *p* ≤ 0.007; stranger duration: F_1, 125_ = 6.546, *p* ≤ 0.01; partner duration: F_1, 126_ = 7.377, *p* ≤ 0.008) and exploratory endpoints (Figure 5A, distance: F_1, 126_ = 10.86, *p* ≤ 0.001; stranger bouts: F_1, 124_ = 9.660, *p* ≤ 0.002; partner bouts: F_1, 120_ = 10.67, *p* ≤ 0.001). A main effect of exposure was only found for the exploratory endpoints (Figure 5A, distance: F_1, 126_ = 9.347, *p* ≤ 0.003; stranger bouts: F_1, 124_ = 9.966, *p* ≤ 0.002). A marginal effect of exposure was found for partner bouts (F_1, 120_ = 2.723, *p* ≤ 0.10). However, there were significant interactions between PD and FM 550 exposure (Figure 4A) on PPI (F_1, 126_ = 4.189, *p* ≤ 0.04), time with the stranger (F_1, 125_ = 4.830, *p* ≤ 0.03) and a weaker, non-significant effect on time spent with the partner (F_1, 126_ = 3.001 *p* ≤ 0.09). Sex and PD were interactive on the three exploratory endpoints with two reaching statistical significance (Figure 5A, distance: F_1, 126_ = 4.970, *p* ≤ 0.03; stranger bouts: F_1, 124_ = 4.804, *p* ≤ 0.03; partner bouts: F_1, 120_ = 3.496, *p* ≤ 0.06).

The social and exploratory PPT data were next examined within sex by two-way ANOVA (Figure 4C and Figure 5B). In females, PD had main effects on all social (Figure 4C) and exploratory (Figure 5B) endpoints assessed, with a small influence of exposure only on stranger duration (Figure 4C). PD females had a lower PPI (Figure 4D, F_1, 59_ = 8.518, η_p_^2^ = 0.13), spent more time with the stranger (Figure 4E, F_1, 58_ = 6.100, *p* ≤ 0.02, η_p_^2^ = 0.10) and less time with their partner (Figure 4F, F_1, 59_ = 8.091, *p* ≤ 0.006, η_p_^2^ = 0.12). PD females also traveled more during the test (Figure 5C, F_1, 59_ = 12.24, *p* ≤ 0.0009, η_p_^2^ = 0.17), and visited both stranger (Figure 5E, F_1, 57_ = 12.20, *p* ≤ 0.0009, η_p_^2^ = 0.18) and partner more often (Figure 5F, F_1, 58_ = 9.036, *p* ≤ 0.004, η_p_^2^ = 0.13). In contrast to females, FM 550 exposure was the predominate main effect on PPT exploratory behavior in males (Figure 5B). The FM 550 males traveled more (Figure 5C, F_1, 67_ = 8.855, *p* ≤ 0.004, η_p_^2^ = 0.12), and made more entries to the chambers with the partner (Figure 5E, F_1, 67_ = 4.830, *p* ≤ 0.03, η_p_^2^ = 0.07) and stranger (Figure 5D, F_1, 67_ = 11.46, *p* ≤ 0.0001, η_p_^2^ = 0.15). An interaction between exposure and PD in males was suggested for time spent with the partner (Figure 4C, F_1, 126_ = 3.001, *p* = 0.09, η_p_^2^ = 0.04). FM 550 apparently countered the negative effect of PD on this behavior (Figure 4F), an effect that was also observed in females.

Within individual groups, the unexposed BPC males (*p* ≤ 0.001) and females (*p* ≤ 0.0001) displayed a partner preference (Figure 4D) as expected, but there was also a sex difference, with females having a higher PPI than males (t_36_ = 2.700, *p* = 0.01, d = 0.90). Unexposed PD females showed a statistically significant partner preference when PPI was calculated for the entire 2-hr test (Figure 4D, *p* ≤ 0.04). However, their PPI was significantly lower than the BPC unexposed females (*p* ≤ 0.005, d = 0.88), with PD females spending more time with the stranger (Figure 4E, *p* ≤ 0.005, d = 0.83) and less time with their partner (Figure 4F, *p* ≤ 0.02, d = 0.79) compared to BPC females. To further investigate PP behavior over the duration of the test, the data were binned into 30-min intervals and analyzed using a two-way ANOVA for pairwise comparisons. Throughout the entire test, the PPI of unexposed PD females was consistently lower (Figure 4G), with more time spent with the stranger (Figure 4H) and less time spent with the partner (Figure 4I) compared to unexposed BPC females. This behavior was particularly apparent during the first hour when unexposed PD females did not display a partner preference (Figure 4G, compared to PP = 0 for equal preference; 30-min: *p* ≤ 0.14, 60-min: *p* ≤ 0.09). Furthermore, PD females traveled (Figure 5F, *p* ≤ 0.01, d = 1.21) and visited the stranger (Figure 5G, *p* ≤ 0.003, d = 1.21) and partner (Figure 5H, *p* ≤ 0.07, d = 0.72) more during the beginning of the test compared to BPC females.

Within females, there was a suggested but not statistically significant, interaction between exposure and PD on stranger duration (Figure 4C, F_1, 58_ = 2.878, *p* = 0.10, η_p_^2^ = 0.05). PD-unexposed females spent more time with the stranger compared to BPC-unexposed females (Figure 4E, *p* ≤ 0.005, d = 0.83) but this effect was not observed in the PD + FM 550 females (*p* = 0.58, d = 0.01). Similarly, PD alone lowered PPI in PD females compared to BPC females (Figure 4D, *p* ≤ 0.005, d = 0.88); however, the combination of FM 550 exposure and lack of paternal care did not (*p* = 0.28, d = 0.50). PD + FM 550 females showed a significant preference for the partner (Figure 4D, *p* ≤ 0.0001). Like the PD-unexposed females, PD + FM 550 females traveled (Figure 5C, *p* ≤ 0.0009, d = 1.14) and visited the stranger (Figure 5D, *p* ≤ 0.002, d =1.53) more than BPC females. Although pairwise comparisons revealed a trend for increased partner bouts in the PD females (Figure 5E, *p* = 0.07, d = 0.72), only the PD + FM 550 group reached statistical significance, suggesting an additive effect (*p* ≤ 0.01, d = 0.87). Furthermore, the PD + FM 550 group traveled more than the BPC FM 550 group (Figure 5C, *p* ≤ 0.02, d = 0.71), which also suggests an additive effect. Pairwise comparisons of binned PP social and exploratory data to BPC females found the PPI of PD + FM 550 was not significantly lower than BPC females at any time point (Figure 4G) such as those observed for PD-only females. Although, PD + FM 550 females traveled more (Figure 5F) with more visits with the stranger (Figure 5G) and partner (Figure 5H) animals than unexposed BPC females during the first hour similar to PD-only females.

In males, FM 550 was the predominant main effect (Figure 5B). Although there was no significant difference between FM 550 males and BPC controls on PPI (Figure 4D), FM 550 exposure altered PP behavior differently across time (Figure 4G). The FM 550 males displayed a weak partner preference overall (Figure 4D, *p* ≤ 0.05), but this was driven mainly by only one time point (Figure 4G, compared PP = 0 for equal preference, 60-min: *p* ≤ 0.01). Unlike the unexposed BPC males, FM 550 male partner preference did not intensify over time (Figure 4G). At 120-min, PPI (Figure 4G) and partner duration (Figure 4I) of FM 550 were significantly lower than BPC males. During the second half of the test, FM 550 males traveled more (Figure 5F) and visited the stranger more (Figure 5G) than BPC males.

Although PD was not found to be a significant main effect on any endpoint (Figure 4C and Figure 5B), PD males lacked a partner preference (compared to PP = 0 for equal preference, *p* = 0.60) and had a significantly lower PPI (Figure 4D, *p* ≤ 0.04, d = 0.68) compared to BPC males. Similar to females, PD males spent more time with the stranger (Figure 4E, *p* ≤ 0.03, d = 0.69) and less time with their partner (Figure 4F, *p* ≤ 0.04, d = 0.72) compared to BPC males. Only the combination of PD and FM 550 significantly altered partner bouts compared to the BPC (Figure 5F, *p* ≤ 0.01, d = 0.80) and PD (Figure 5F, *p* ≤ 0.02, d = 0.80) controls, suggesting the combination uniquely heightened this behavior in males. By contrast, compared to the BPC group, PD elevated time spent with the stranger in males (Figure 4E, *p* ≤ 0.03, d = 0.69), an effect not found in the PD + FM 550 group (*p* = 0.65, d = 0.16), suggesting exposure countered the effect of PD. Similar to BPC FM 550 males, PD + FM 550 males displayed a weak partner preference (Figure 4D, *p* ≤ 0.02), mostly accounted for by one time point (Figure 4G, 90 min, *p* ≤ 0.01) with PPI not intensifying over time.

## 4. Discussion

This study expands on our previous work examining the effects of perinatal FM 550 exposure on socio-emotional behavior by incorporating an ELS. As expected, PD adversely impacted behavior, with increased anxiety, decreased sociability, and impaired pair-bonding observed in both sexes, particularly females. As we saw twice previously using prairie voles [57,58], FM 550 heightened anxiety and partner preference in females but reduced partner preference in males. The combination of FM 550 exposure and PD did not exacerbate any behaviors in either sex, except for distance traveled by females in the PPT and, to a lesser extent, time spent with, and the number of visits to the empty cup by males in the ST. Contrary to the hypothesis, FM 550 ameliorated the impacts of parental deprivation on partner preference behaviors in both sexes. We previously asserted that the higher anxiety displayed by FM 550-exposed females makes them more eager to seek familiarity as a comfort strategy and, consequently, spend more time with their partners and avoid the unfamiliar. The present study results are concordant with that prior work, with a similar phenomenon happening, perhaps, even in females that also experienced the ELS. In males, the data suggest it is more likely that FM 550 heightens exploratory drive and novelty-seeking, rather than anxiety, which reduces PPI. This study is significant because it provides evidence that a combination of chemical and social stressors can have unique effects that differ by sex.

As expected, FM 550 exposure and PD independently heightened anxiety-like behavior in females, as shown by decreased entries and time spent in the center of the OF arena. Although neither effect quite reached statistical significance, the outcomes are consistent with what we observed previously in both rats and voles using OF and other tests of anxiety-related traits [55,57,59]. Additive aversive effects would be difficult to observe due to a “floor” effect (the center activity cannot go any lower). The male data revealed different effects of PD and FM 550, with no evidence of a combined influence on any OF endpoint. PD produced anxiogenic effects on latency to enter the center and the number of center entries, while FM 550 heightened distance traveled and time in the center, effects more consistent with heightened exploratory behavior rather than a change in anxiety [56,59]. Notably, in both sexes, as expected, exploration of the arena decreased as the test progressed, indicating all animals habituated to the arena. Thus, aversion was specific to the center and no group habituated more slowly than any other.

The ST assesses both social and exploratory motivation, and we previously showed that developmental FM 550 exposure impairs aspects of sociability in this test in prairie voles of both sexes [57]. However, in the present study, PD was more impactful. While a potentially synergistic effect was observed in PD + FM 550 exposed males with a perceivable decrease in sociability, this weak effect did not reach statistical significance. This was accompanied by increased interaction with the empty cup. To more robustly disentangle behaviors related to novelty from those related to sociability, alternative social preference tasks that use, for example, a same-sex familiar animal versus a same-sex stranger animal may yield greater insight as to effects on social motivation. Similarly, novel object tasks and related mazes would more robustly delineate exposure-related effects on exploratory behavior, novelty-seeking, and aspects of spatial memory. Additionally, future studies will be necessary to probe for FM 550-related effects on fear and fear-mediated behaviors, as heightened fear may influence these pathways more directly than sociability, especially in females. In that regard, it is possible that FM 550 and PD impact different socially impactful, but not overlapping, pathways and thus why the two in combination were largely not synergistic.

Partner preference was the most profoundly impacted behavior with both stressors altering behavior individually and in combination. The outcomes were heavily influenced by sex, which was not unanticipated given the strongly sexually dimorphic nature of this behavior and its neuroendocrine systems. As others have repeatedly shown [71,73,77,94,95], lack of paternal care adversely impacted pair-bonding with a lack of partner preference observed in both sexes. Moreover, this is the third time that we showed that developmental FM 550 exposure sex-specifically alters pair-bonding [57,58]. As in our previous studies, FM 550 females displayed and maintained a strong partner preference throughout the task, while exposed males failed to consistently show a preference for their partner. Notably, preference decreased in the second half of the test in both the FM 550 and PD groups, contrasting with the unexposed males. Whether or not the FM 550 males would mate with a stranger female, given the opportunity, remains to be determined. If anything, in the combined exposure males, FM 550 ameliorated the affiliative deficits of PD to some degree. The mechanisms by which this occurs are unknown but could indicate FM 550 increases novelty-seeking in males.

Interestingly, females exposed to the combined stressors displayed stronger affiliation with their partners than PD females. This is consistent with our prior conclusion that FM 550 exposed females have high generalized anxiety and remain close to familiar settings and animals, especially their partners, as a coping/calming strategy. This effect seems to override the compromising influence of ELS on partner preference behavior. That proximity to the partner can be an effective coping strategy in this species was demonstrated in prior work. For example, female prairie voles exposed to 1-hr immobilization stress show elevated anxiety in recovery, but not if allowed to recover with their male partner [96]. This hypothesis will be the focus of future studies.

The mechanisms by which PD alters social behaviors in each sex are well-understood, while those by which FM 550 exposure alters similar behaviors are almost totally unknown. Oxytocin (OT), vasopressin (AVP), and mesolimbic dopamine pathways are integral to the promotion of pair-bonding and affiliative behaviors [97,98,99]. The density of OT and AVP receptors in key brain regions strongly influences the display of monogamous behaviors. In monogamous vole species, males have a greater density of the AVP 1a receptor (AVPR1a) in the lateral septum and ventral pallidum, while females have higher OT receptor (OTR) expression in the nucleus accumbens compared to promiscuous species including mice and other vole species. Dopamine receptors in the nucleus accumbens promote a rewarding effect of pair formation and maintenance. The paraventricular nucleus (PVN) and amygdala are integrative brain regions involved in sociality, stress responses, and anxiety. Generally, AVP has anxiogenic but OT anxiolytic effects [100]. Studies in mandarin voles, which are also socially monogamous, have shown that PD decreases the number of OT neurons originating from the PVN and projecting to the medial prefrontal cortex in females [101], while PD male mandarin voles have more AVP-immunoreactive PVN neurons and fewer AVP-immunoreactive neurons in the anterior hypothalamus and fewer OT-immunoreactive neurons in the PVN [102]. In PD prairie voles, female OTR mRNA expression is lower in the medial amygdala and nucleus accumbens [103] while AVPR1a density is higher in the medial amygdala [76], which is consistent with reduced partner preference and higher anxiety. In male prairie voles, PD was shown to reduce OTR density in the central amygdala [76].

The mechanisms by which FM 550 exposure alters socio-emotional behavior likely involve OT, AVP and the mesolimbic dopamine system. We previously showed that developmental FM 550 exposure decreases PVN AVP neuron numbers in female prairie voles [54]. However, we found no effect in either sex on PVN OT neuron numbers or compelling evidence that the density of dopaminergic neurons is altered in related areas. Although, we found developmental exposure alters electrophysiological properties in the nucleus accumbens in both sexes [58], suggesting the possibility that dopaminergic activity is altered in this key brain region for monogamy. Since it is not the density of OT and AVP neurons, but rather, their receptors that drive monogamous behaviors, studies already underway are assessing OT and AVP receptor levels in multiple brain regions, including the nucleus accumbens.

## 5. Conclusions

This study again demonstrates that prairie voles are an exemplary model organism for exploring the impacts of chemical exposure and other external stressors on social behavior because they are spontaneously affiliative. Toxicological rodent studies have traditionally been conducted in rats, mice and other species that do not display monogamy, paternal care, or strongly affiliative behaviors. Yet humans are strongly prosocial, thus, the prairie vole is a more valuable translational species for toxicology studies probing for adverse effects on social traits and related neuroendocrine systems. Significantly, we observed altered pair-bonding following FM 550 exposure in this unique species in three, independently conducted studies that indicate that the phenotypes are highly reproducible. Using different FM 550 doses and testing the effects of the individual components is an obvious next step, as is the use of alternative tests to more comprehensively assess exploration, fear, sociality, and novelty-seeking. The mechanisms by which the observed socio-emotional effects occur likely involve OT and AVP systems and their intersections with the mesolimbic dopamine system and the stress axis, but they have not yet been elucidated. These and other mechanistic hypotheses are in the process of being explored.

## Figures and Tables

**Figure 1 toxics-10-00268-f001:**
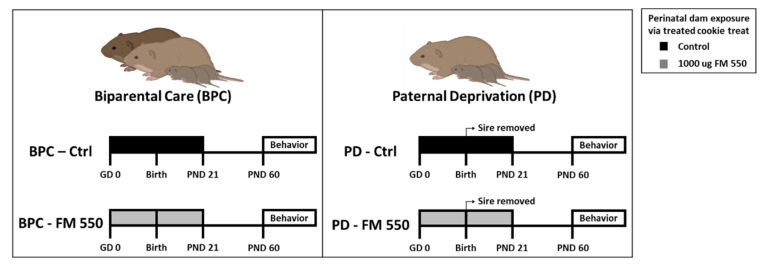
Study design.

**Figure 2 toxics-10-00268-f002:**
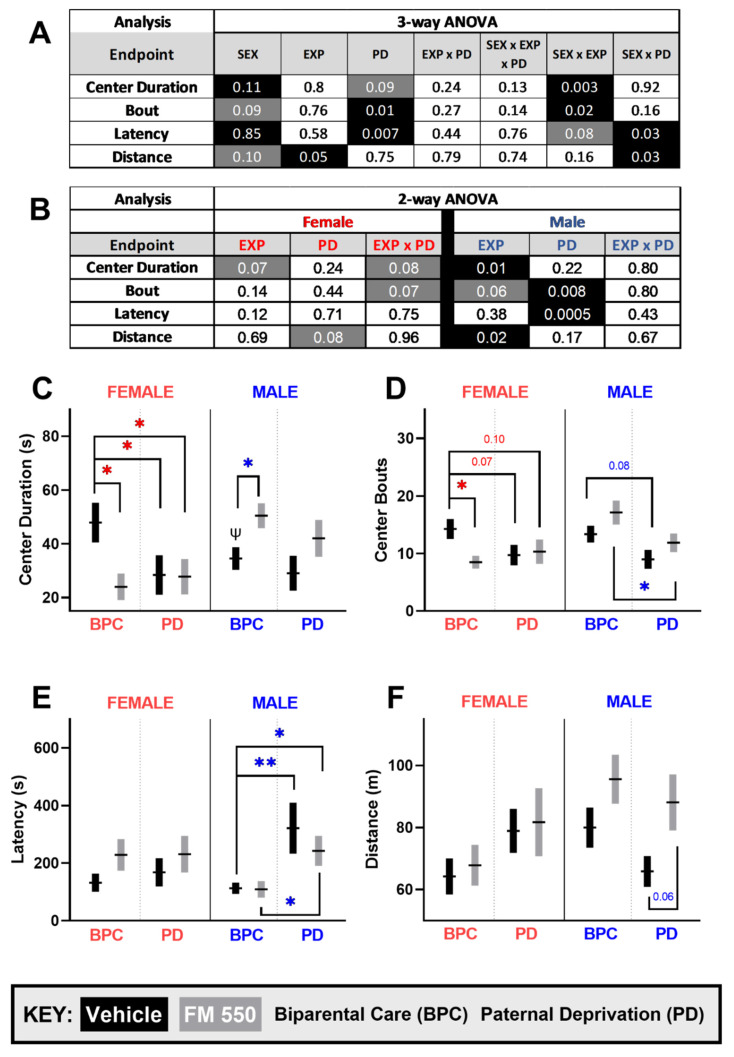
Open-field outcomes. (**A**) Three-way ANOVA *p*-values for main effects of sex, FM 550 exposure, and paternal care and (**B**) Two-way ANOVA within sex *p*-values for each endpoint. Significant effects are highlighted in black and suggestive effects (*p* ≤ 0.10) are highlighted in grey. (**C**) FM 550 females spent less time and made fewer entries (**D**) in the center than unexposed females. (**C**) FM 550 males spent more time in the center than the unexposed males. Both the unexposed and exposed PD females spent less time in the center (**C**) with a suggestive but not significant decrease in center entries (**D**) than the BPC unexposed females. (**E**) Similarly, unexposed and exposed paternally deprived males took longer to enter the center than BPC unexposed males. (**F**) The main effect of FM 550 was found for males on distance traveled. Graphs depict mean ± SEM. For each dose (*n* = 13–25), * denotes statistically significant difference between groups within sex, while ψ denotes significant sex differences between BPC controls. A single symbol represents *p*^(^*^,*ψ*)^ ≤ 0.05 and a double symbol represents *p*^(^**^)^ ≤ 0.01.

**Figure 3 toxics-10-00268-f003:**
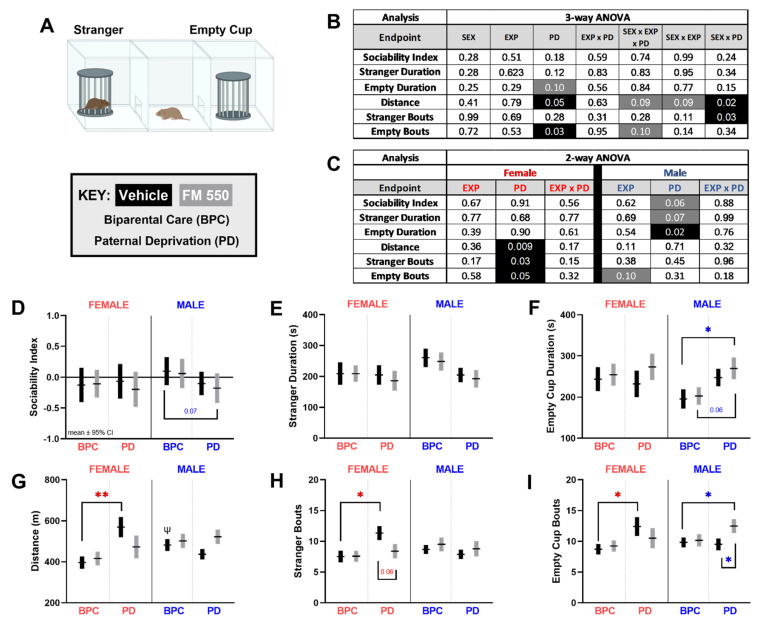
Sociability test. (**A**) Arena schematic depicting placement of each element. (**B**) Three-way ANOVA *p*-values for main effects of sex, FM 550 exposure, and paternal care and (**C**) Two-way ANOVA *p*-values for each endpoint. Significant effects are highlighted in black and suggestive effects (*p* ≤ 0.10) are highlighted in grey. Main effects were primarily driven by paternal deprivation (**B**) and primarily in females on the exploratory endpoints in the task (**C**). The sociability index revealed no preference for either cup in any group, however, a suggested lower sociability index PD + FM 550 males compared to BPC males (**D**). No significant differences in duration with strangers were found in either sex (**E**) however, there was a suggestive paternal care effect in males. PD females traveled more (**G**) with more entries with the stranger (**H**) and empty cup (**I**) than BPC females. PD + FM 550 males visited (**I**) and spent more time with the empty cup than BPC males (**F**). Graphs (**E**–**I**) depict mean ± SEM and (**D**) depicts mean ± 95% CI. For each dose (*n* = 15–28); * denotes a statistically significant difference between groups within sex, while ψ denotes significant sex differences between BPC controls. A single symbol represents *p*^(^*^,*ψ*)^ ≤ 0.05 and a double symbol represents *p*^(^**^)^ ≤ 0.01. For the sociability index, a significant difference from chance (SI = 0, solid line). A sociability index of 1.0 indicates preference for stranger animal and an index of −1.0 indicates preference for the empty cup.

**Figure 4 toxics-10-00268-f004:**
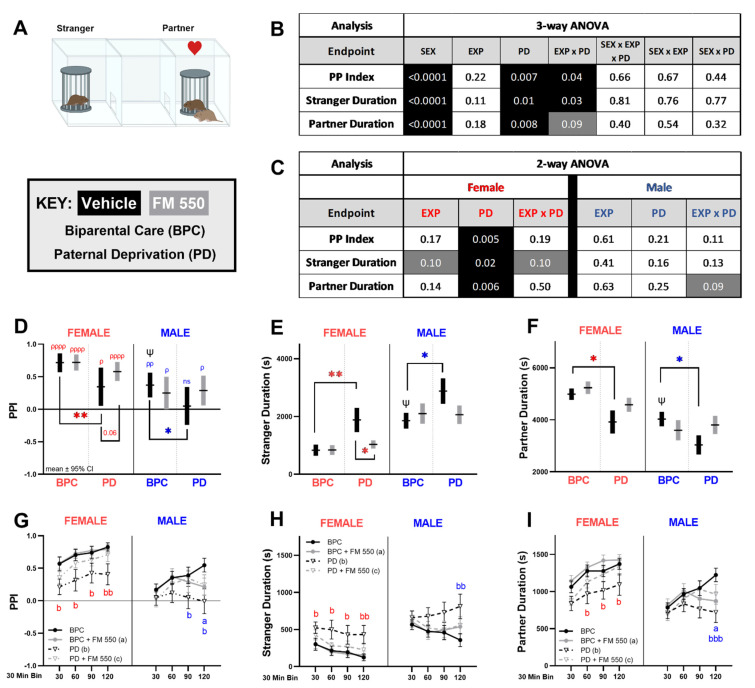
Social endpoints in the partner preference test. (**A**) Schematic depicting the placement of the partner and stranger animals in the three-chamber arena. (**B**) Three-way ANOVA *p*-values for main effects of sex, exposure, paternal deprivation, and interaction for each endpoint. Significant effects are highlighted in black and suggestive effects (*p* ≤ 0.10) are highlighted in grey. (**C**) Two-way ANOVA *p*-values within sex. (**D**) Partner preference index, calculated over the entire 2 hrs, was sexually dimorphic and paternal deprivation significantly lowered PPI in both unexposed groups (male and female). Only the PD males did not show a partner preference. (**E**) Time spent with the stranger was sexually dimorphic and PD females and males spent more time with the stranger than their BPC counterparts. FM 550 PD females spent significantly less time with the stranger than PD only females. Similar findings were found for partner duration (**F**), with PD males and females spending less time with their partners than BPC controls of the same sex. Endpoints were also binned into 30-min intervals (**G**–**I**) to explore PP behavior over time. Notably, PPI and partner duration tended to increase with time in the BPC groups, but this pattern was not seen in the PD males. Graphs (**E**–**I**) depict mean ± SEM and (**D**) depicts mean ± 95% CI. For each dose (*n* = 14–23), * denotes statistically significant exposure effects within sex, while ψ denotes significant sex differences between unexposed BPC animals. For PPI (**D**), a significant difference from equal preference (PPI = 0, solid line) is indicated by ρ. A lack of significant difference is indicated by ns. A partner preference index of 1.0 indicates a maximal preference for a partner, while an index of −1.0 indicates a maximal preference for the stranger. For binned data (**G**–**I**), significant group differences at individual time points are depicted by letters (a: BPC + FM 550; b: PD; c: PD + FM 550); circles with solid lines = BPC, triangles with dashed lines = PD, black = unexposed, grey = FM 550 exposed. A single symbol represents *p*^(^*^,*ρ*,*ψ*,*a*,*b*)^ ≤ 0.05, a double symbol represents *p*^(^**^,*ρρ*,*bb*)^ ≤ 0.01, a triple symbol represents *p*^(*bbb*)^ ≤ 0.001, and four symbols represent *p*^(*ρρρρ*)^ ≤ 0.0001.

**Figure 5 toxics-10-00268-f005:**
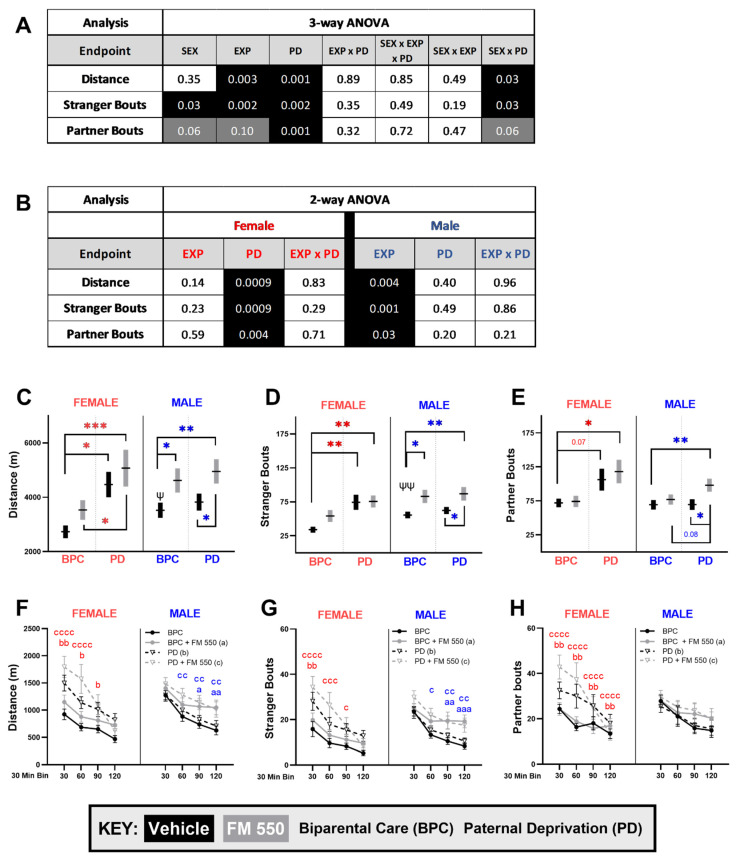
PP exploratory behavior in the partner preference test. (**A**) Three-way ANOVA *p*-values for main effects of sex, exposure, paternal deprivation, and interaction for each endpoint. Significant effects are highlighted in black and suggestive effects (*p* ≤ 0.10) are highlighted in grey. (**B**) Two-way ANOVA *p*-values within sex. Main effects of exposure were only seen in males, while the main effects of PD were only observed in females. (**C**) Effects on distance traveled were influenced by paternal care in females but exposure in males, with co-exposed females traveling more than any other female group. Similar findings were found for bouts with the stranger (**D**) and partner (**E**) animals, where a main effect of paternal care was found in females but exposure in males. Endpoints were also binned into 30-min intervals (**F**–**H**) to explore PP behavior over time. Overall activity decreased over time as the animals habituated to the task. PD females were more active early in the task, while FM 550 exposed male activity was higher than unexposed male activity towards the end. Graphs (**C**–**H**) depict mean ± SEM. For each dose (*n* = 14–23), * denotes statistically significant exposure effects within sex, while ψ denotes significant sex differences between unexposed BPC animals. For binned data (**F**–**H**), significant group differences at individual time points are depicted by letters (a: BPC + FM 550; b: PD; c: PD + FM 550); circles with solid lines = BPC, triangles with dashed lines = PD, black = unexposed, grey = FM 550 exposed. A single symbol represents *p*^(^*^,*ψ*,*a*,*b*,*c*)^ ≤ 0.05, a double symbol represents *p*^(^**^,*ψψ*,*aa*,*bb*,*cc*)^ ≤ 0.01, a triple symbol represents *p*^(^***^,*aaa*,*ccc*)^ ≤ 0.001, and four symbols represent *p*^(*cccc*)^ ≤ 0.0001.

## Data Availability

Not applicable.

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
