# Peer review of "Individual and Combined Effects of Paternal Deprivation and Developmental Exposure to Firemaster 550 on Socio-Emotional Behavior in Prairie Voles"

_toxics, 2022, doi:10.3390/toxics10050268_

Round 1
Reviewer 2 Report
Gillera et al. have conducted an important study focused on the effect of a chemical flame retardant mixture, Firemaster 550, and stress on social behavior. One important advance in this study is the use of the prairie vole, as their bi-parental care creates a higher ceiling for the identification of social deficits. These rodents show a wide range of social behaviors that may be more informative for the influence of chemical mixtures and their interactions (or lack thereof) with non-toxicant stressors.
Early life stress is an important and complicated issue because different stress paradigms trigger different responses. In prairie voles, paternal deprivation is clearly an environment with reduced social interaction, altered maternal behavior, and maybe elevated glucocorticoids (although these data were not provided). I would be more cautious in equating this experience with the experience of children raised with single mothers. This is a complicated experience that cannot be easily reduced to presence or absence of father. I would explicitly note that issues of parental absence and social support are more complicated in humans. If anything this is a model of social instability or reduced social contact broadly, more than it is a direct reflection of paternal absence.
Can you measure the corticosterone in the pups at weaning or maybe a few days after the males is removed? How much of these effects come from HPA axis stimulation?
Novelty stress versus reduced parental support: Do females habituate to the lack of a male in the cage? Have you measured maternal behavior or CORT in the dams? Fathers are removed at birth resulting in a novel acute environmental change. Is there any habituation to his absence?
Litter effects: Just for clarification: “No more than 4 same sex pups per litter were used in a single endpoint, no more than 2 total litters and a maximum of 5 total pups per sex were selected per mated pair.” I don’t understand this completely. Can you clarify the animal numbers by litter?
Which treatment groups were excluded due to the loss of 21 videos in OF?
With this large number of animals how were animals counterbalanced across behavioral testing? Were they tested with all treatments and sexes randomized?
How many times were the “strangers” used? How was the behavioral experience of the stranger accounted for?
Prairie voles can take longer to habituate to behavioral arenas (especially when tested individually because you get this separation anxiety). In your open field data, do you see expected habituation patterns (reduced activity across the 30 minutes) as observed in inbred mice if you look at the data in 5 minute bins across the session? Do you think individual rates of habituation to the open field influenced increased distance in FM 550 males? Did they have hyperactivity across the whole session (there was a main effect of exposure on distance)?
In the methods for the sociability test, lines 239 to 241 state that voles are able to explore a stranger or novel object, but then then next line suggests the alternative cup was empty. Was it empty or did it have a novel object?
While I generally support discussions of values below 0.10 as marginally significant or marginally not significant, these studies have a very large number of animals. There should be some clarity in the discussion and more care used for the non-significant changes. Currently, all changes are being discussed with a similar magnitude. An example of a sentence that needs to change is “A synergistic effect 522 was observed in PD FM 550 exposed males with a perceivable decrease in sociability, albeit this was a weak effect that did not reach statistical significance.” This was not significant and it is a very big claim that the effect was synergistic.
One final note, the data tables are perfect! I am going to start using these myself to clarify when I have a main effect versus interaction versus when I am reporting post-hoc tests. This is very hard to do well. It can get very complicated. In the two-way ANOVAs, (separating by sex is so important), There are no significant interactions. This is important. Throughout the manuscript, the effects of the post-hoc tests are discussed but the overall main effect is downplayed.
The evidence of enhanced effects is limited. This is discussed, however, what if these two insults simply do not share a direct underlying biological substrate? Perhaps it is interesting enough that there were not synergistic effects seen. Maybe the effect of FM 550 targets fear mediated pathways, more than selectively targeting social bonding? This is worth discussing as an alternative.
I agree that the reduced effect in stranger duration in the FM 550-PD females compared to the control-PD females can be associated with increased “anxiety-like” behavior and/or fear of novelty. Contrary to the interpretation that FM 550-PD females are spending more time with their partner to cope with this "anxiety", it seems that they have elevated distance traveled compared to FM-550-BPC females and (although not significant) lower partner duration compared to the BPC females (control and FM 550). I make the second claim because, the main effect of Parental care is significant for reduced partner duration. The data do not support that they are using the partner to cope. FM 550-PD females show equal bout numbers with the strangers compared to the control-PD females, they must just have shorter bout durations to account for the total reduction in stranger duration (4E). This is concordant with increased anxiety-like behavior or altered exploration of the stranger, but I am not sure this is evidence that they are using the familiar partner to cope. They simply seem hesitant to investigate strangers for longer durations, with increased distance traveled and some evidence of reduced partner duration. I am happy to discuss more.
Reviewer 3 Report
Manuscript Number: toxics-1659034
Title: Individual and combined effects of paternal deprivation and developmental exposure to Firemaster 550 and Socioemotional Behavior in Prairie Voles
Authors: S.E.A. Gillera, W. P. Marinello, M.A. Nelson, B. M. Horman, and H.B. Patisaul
The authors present data on interaction of maternal flame retardant exposure and neonatal paternal deprivation, an early life stress, on adult offspring behaviors – avoidance, social, and partner preference –in a prosocial rodent species, the prairie vole. This document was well written and easily read and understood by the reviewer. A thorough explanation of the experimental design and the caveats to the design is provided and the statistical analyses of the complex multi-factorial design are described in detail. The discussion sufficiently reviews the literature – largely by the senior investigator’s lab – and offers potential avenues of investigation for the underlying mechanism for both flame retardant exposure and paternal deprivation.
Major Comments:
Methods – justification was given for not using the dam as the statistical unit. However, the authors should clarify what they mean by “that potential confound was handled statistically”. I do not see any discussion in 2.8 about analyzing for dam or the confounding influence of dam exposure.
Results – two figures are difficult to read as laid out in the reviewing format. Figures 2 and 4 should be flipped to landscape and made larger or divided (Figure 4) into two figures. The reader/reviewer cannot read the x- or y-axis information or legends. While the Results text described the data sufficiently, the reviewer/reader cannot understand the data by just by looking at the figures.
Discussion – is there any data indicating that the influence of paternal deprivation is due to the stressors on the dam rather than the stressors on the pup? If mate separation causes the dam to be stressed then I would imagine the changes to maternal care is an equal or even greater influence than the lack of paternal care itself.
Minor Comments:
Methods – Page 4, lines 177-178 – add ‘in’ after ‘prior work’ – FM 500 doses were selected based on our prior work in voles in which FM 550 exposure….
Results – Delete “This section may be divided by subheadings. It should provide a concise and precise description of the experimental results, their interpretation, as well as the experimental 298 conclusions that can be drawn.” That appears to be instructions to the authors.
Results – Page 11, lines 439-440 – place a semi-colon and comma before and after however “….d = 0.88); however, the….”
Reviewer 4 Report
The manuscript describes that exposure to FM550, a flame retardant mixture, increases some components of anxiety and influences partner preference in the offspring. It is also presented that paternal deprivation increases anxiety and impairs pair bonding in the offspring. While the combination of FM550 exposure and paternal deprivation exhibits unique effects that differ by sex. Nevertheless, initial enthusiasm from this study is somewhat overlapped with a complicated introduction and poorly explained rationale. The manuscript is generally written, some issues need to be clarified.
- The manuscript is in need of writing polishing, as many grammar issues plague it and reduce the impact of the reading.
- When describing the behavioral tests, what is the sequence of three behavioral tests and how long is the latency between each test? It could be possible that shorter inter-test interval may be not enough to “washout” the effect of previous test, especially on exploring the anxiety and social interaction.
- In Figure 2A and Figure 2C, the “center duration” parameters shown in figures are different.
- In Figure 3G, it is unnecessary to show “ns” above each group.
- In the abstract is necessary to more clearly formulate the main results of the study.
Round 2
Reviewer 2 Report
Thank you for your responses. Most responses were very detailed and added clarity. Please clarify a few follow-up questions.
Litter effects: Just for clarification: “No more than 4 same sex pups per litter were used in a single endpoint, no more than 2 total litters and a maximum of 5 total pups per sex were selected per mated pair.” I don’t understand this completely. Can you clarify the animal numbers by litter?
Because voles mate for life and litter sizes are small in some cases pups were taken from sequential litters of the same pairs. A maximum of 2 litters were used from each breeding pair. The maximum number of same sex pups from each breeding pair (total, across both litters) was 5. We also amended the methods to make this section a little more clear.
Response: If multiple litters per pair were used, how are the males returned to the cage for the next litter to be born?
On line 210, it is noted “that potential confound (of litter) was handled statistically by testing for litter effects to which none were identified”. This is a great. Can the author’s include supplemental tables showing that the litter was not significantly different. This will help highlight that moving the statistical unit to the pup was not problematic.
With this large number of animals how were animals counterbalanced across behavioral testing?
The animals were tested in random order and as they became available. Because the experimenter was blind to exposure (but not sex), and the animals were generated over months, test order was entirely random.
Response: Blinding is very important. Now that the data are un-blinded, were the animals evenly distributed by time of day or cohort? Such that, no one group was always tested in the morning and one group always tested in the afternoon, etc. Chance suggests they are evenly distributed, but can you double check the effect of time of day?
I did not realize that the animals were tested in a cohort design (over months as they became available). Please add this information to the methods, date range of testing and the number of cohorts. Was there any statistically significant effect of cohort? Please also note that the males were always tested first (in the morning?) and females second in the methods.
